

# Optimal allocation of 30GW offshore wind power in the Norwegian Economic Zone

Sondre Hølleland[1], Geir Drage Berentsen[1], Håkon Otneim[1], and Ida Marie Solbrekke[2]

[1]Norwegian School of Economics, Helleveien 30, 5045 Bergen, Norway
[2]Norwegian Research Centre (NORCE), Jahnebakken 5, 5007 Bergen, Norway
**Correspondence:** Sondre Hølleland (sondre.holleland@nhh.no)

**Abstract.** The Norwegian government aims to install offshore wind power with a total capacity of 30 gigawatts by 2040, and the Norwegian Water Resources and Energy Directorate has suggested twenty candidate regions. We show that the potential for reducing overall power production variance across these regions is high using modern portfolio theory and the hourly and spatially rich reanalysis NORA3 Wind Power data set (NORA3-WP). The geographical diversification effect is demonstrated

under various relevant scenarios, including a sequential build-out scenario with a fully connected Norwegian power grid assumption. By considering 20 alternative regions selected using a recently developed suitability score, we further illustrate that the diversification effect is robust to location changes.

## 1   Introduction

Policymakers, environmental organizations, industry, and researchers portray offshore wind power as a vital energy source

to meet the increasing demand for clean, renewable energy as the world transitions from fossil fuels. Norway has a large and largely unexploited potential for offshore wind power production (Bosch et al., 2018). The Norwegian Government has presented an ambitious development plan, called "30by40", of continuously opening offshore areas for large-scale wind power deployment, sufficient for 30 gigawatts (GW) installed capacity by 2040 (Norwegian Government, 2022). Theoretically, having only one wind farm with 30 GW of installed capacity would occupy 9400 km$^2$, corresponding to a square with sides of 97 km

(Solbrekke, 2022). However, distributing wind farms across a larger geographic area could stabilize the instantaneous power production (Solbrekke et al., 2020; St.Martin et al., 2015). If there is little wind in one area, this can be compensated by windy conditions somewhere else. This effect is analogous to diversification in financial portfolio selection problems using modern portfolio theory (Markowitz, 1952). Thus, we consider the distribution of wind farms as an optimization problem, where we aim to maximize power production while minimizing its variance. In the context of opening several areas for offshore wind

power deployment, where the instantaneous wind resources are more or less dependent, it is crucial to first determine the location of potential wind farms. We then apply modern portfolio theory to determine the relative sizes of the wind farms to obtain the best tradeoff between power output and stability. Our objective is to find the portfolio that minimizes the variance given a specific expected power output. Modern portfolio theory has a long history in financial portfolio selection, where the weights represent how large a portion of the total investment one should invest in different stocks, bonds, or funds. In a wind





farm portfolio, the weights correspond to the proportion of the total number of wind turbines potentially installed at each wind farm location.

The Norweigan (exclusive) economic zone (NEZ) is extensive, and not all areas are suitable for installing turbines. In most areas, the sea depth is too large for anchoring wind turbines to the sea floor at an acceptable cost. Some areas are known to be spawning grounds for fish, while other areas may be too close to other offshore installations, such as oil and gas platforms. In this paper, we only consider locations that are suitable for the installation of offshore wind turbines. We achieve this by using two different approaches when selecting suitable candidate sites. Our first set of candidate locations was suggested by The Norwegian Water Resources and Energy Directorate (NVE). NVE suggests 20 areas in the NEZ for further consideration in a subsequent impact assessment. Our second set of candidate locations is based on the study by Solbrekke and Sorteberg (2023), who use multicriteria decision analysis to point out suitable and robust offshore areas for wind power deployment.

Markowitz's modern portfolio theory has been applied to wind power production several times in the literature. Drake and Hubacek (2007) study the geographic diversification effect of wind farm portfolios in the United Kingdom by comparing a portfolio of 2.7GW in one location to one where the same energy is distributed over four locations. They find a reduction of 36% in the standard deviation of instantaneous wind power production. Roques et al. (2010) consider total wind production data from five European countries (Spain, France, Germany, Denmark, and Austria) and apply modern portfolio theory to minimize variance, in a theoretical unconstrained portfolio as well as a portfolio in which national wind resource potential and transmission constraints are taken into account. Rombauts et al. (2011) build on the work of Roques et al. (2010), but instead of using aggregated data by country, they apply portfolio theory on simulated data from different locations within each country. They also model cross-country transmission constraints more explicitly. With case studies from the United States, Degeilh and Singh (2011) propose a general planning method to minimize the variance of aggregated wind power output by optimally distributing turbines over a preselected number of potential sites, Novacheck and Johnson (2017) study the potential for diversification of wind power variability in the Midwest, and Costa-Silva et al. (2017) use modern portfolio theory with four re-balancing strategies on 11 hypothetical offshore wind farms off the East Coast. More recently, Tejeda et al. (2018) employed the ERA-Interim wind resource reanalysis data to minimize the variability of aggregated wind farm production over a $0.25 \times 0.25$ degrees grid of onshore Europe (EU-28) and a selection of offshore grid cells. Finally, Hjelmeland and Nøland (2023) analyse the correlation structure between potential Norwegian offshore wind resources and existing resources in neighbouring countries concerning potential price effects.

In this study, we use the high-resolution wind power reanalysis NORA3-WP (see section 2.1) for our analysis. In contrast with ERA-Interim, which Tejeda et al. (2018) employed, NORA3-WP has a higher temporal resolution (hourly versus 6-hourly) and a longer history (24 years versus 10 years). Furthermore, the analysis by Tejeda et al. (2018) includes most of the European continent, while we focus exclusively on sites in the NEZ suitable for offshore wind power installations. Instead of placing wind power by grid cell (Tejeda et al., 2018), we find the optimal number of turbines on a wind farm unit represented by the wind resources from one grid cell in NORA3-WP. Our study, therefore, builds naturally on recent developments in the identification of suitable locations for wind power generation (Solbrekke and Sorteberg, 2023) and methodology for distributing resources between them (Tejeda et al., 2018). We develop the procedure for wind power distribution further by incorporating





the Norwegian government's sequential development plan, "30by40", and introducing a maximum number of wind farms constraint (see Section 3).

The structure of this paper is as follows. We present the NORA3-WP data in Section 2.1. We then present the NVE candidate locations and the Solbrekke and Sorteberg (2023) counterpart. In Section 3, we present Markowitz's portfolio theory with particular adaptations specific to the wind power problem. We set up five cases with varying constraints and build-out strategies.

The optimal portfolios are presented and discussed in Section 4. We then give some concluding remarks in Section 5.

## 2    Data

### 2.1    NORA3-WP

The backbone of this study is the new wind resource and wind power data set NORA3-WP constructed by Solbrekke and Sorteberg (2022) and validated against observations from offshore installations (Solbrekke et al., 2021). It is based on the 3-km

Norwegian reanalysis data (NORA3), which is the most recent high-resolution data archive from the Norwegian Meteorological Institute (Haakenstad et al., 2021), generated by a dynamical down-scaling of ERA5 (Hersbach et al., 2020). NORA3-WP covers the North Sea, the Norwegian Sea, the Baltic Sea and parts of the Barents Sea in a 3km×3km horizontal grid. NORA3-WP contains climatological data on a monthly time scale from 1996 to 2019, providing seven wind resources and 18 wind power-related variables for three selected turbines with different power ratings, turbine diameters and hub heights. The under-

lying hourly wind speed and wind power data are also available on the same spatial scale. For more details on the data, see Solbrekke and Sorteberg (2022).

In this study, we use the hourly wind power data, calculated using the 15MW reference turbine from the International Energy Agency, IEA-15MW (Gaertner et al., 2020), which is the largest among the three turbines covered by NORA3-WP. IEA-15MW has a rated power of 15MW, a hub height of 150 meters and a rotor diameter of 240 meters. If the wind speed is below 3.0 m/s

or above 25 m/s the power production is zero due to internal friction and sheltering purposes, respectively. If the wind speed is between 3.0 m/s and 10.59 m/s, the power production is proportional to the wind speed cubed. Lastly, if the wind speed lies between 10.59 m/s and 25 m/s, the turbine produces its rated power.

Using the IEA-15MW turbine in our analysis implies that installing 30GW of offshore wind power corresponds to building 2 000 turbines. From the NORA3-WP data, we extract hourly time series from the grid cells closest to the centre of the NVE

regions described in the next section, and the actual grid cells from the Solbrekke and Sorteberg (2023) selected locations in the following section. We calculate the mean capacity factor, i.e. the average production as a percentage of rated power, and the covariance matrix describing the linear dependence between the different locations. This mean vector and covariance matrix is then used in the portfolio optimization described in Section 3. Since the capacity factor is a relative measure in $[0, 1]$, we report it as a percentage. Note that the standard deviation of a capacity factor is then measured in percentage points (pp).





## 2.2 Norwegian Water Resources and Energy Directorate (2023) candidate locations

The Norwegian Water Resources and Energy Directorate (NVE) led a group with members from different state agencies (Norwegian Directorates for Petroleum, Fisheries, Environment and the Coastal Administration and Defence Estates Agency) with a mandate from the Norwegian Ministry of Petroleum and Energy to identify suitable locations for offshore wind farms that have few conflicting interests (Norwegian Water Resources and Energy Directorate, 2023). The group of directorates identified 20 regions suitable for wind power (see shaded areas in Figure 1D). In September 2023, the Norwegian government instructed NVE to start an impact assessment on three of these 20 areas: *Vestavind B, Vestavind F* and *Sørvest F* (Norwegian Government, 2023). Parts of Sørvest F and Vestavind F have earlier been considered for wind power production under the names Sørlige Nordsjø 2 (SN2) and Utsira Nord (UN), respectively. SN2 and UN were identified in an earlier report by NVE (Norwegian Water Resources and Energy Directorate, 2012), and it has been decided to allocate areas for 1500MW at each location to start with. We will, therefore, make sure these are in both candidate sets and use the names SN2 and UN for the corresponding areas in both sets. The suggested regions from 2012 were also used in the analysis by Hjelmeland and Nøland (2023). The NVE used a tool called "Marine Resources Tools", or MaRS, for selecting the newest areas (The Crown Estate, 2019). This is essentially a suitability analysis that excludes certain areas due to input from the interest group members.

The total area covered by the 20 regions suggested by NVE is $54\,867$ km$^2$. Norwegian Water Resources and Energy Directorate (2023) uses different capacity densities (3.5, 5 and 7.5 MW/km$^2$) and different area utilization rates (33%, 67% and 100%). We choose the lowest capacity density and a 100% utilization rate for our study[1]. Using these parameters, we can calculate the maximum number of turbines per region by

$$\text{Potentially installed capacity} = \text{Area} \cdot 3.5\,\text{MW/km}^2 \cdot 100\%, \quad \text{Maximum NoT} = \left\lfloor \frac{\text{Potentially installed capacity}}{15\,\text{MW}} \right\rfloor,$$

where $\lfloor \cdot \rfloor$ means rounding down to the nearest integer, and NoT is the number of turbines. The area and resulting potential capacity and maximum number of turbines per region are given in Table 1. The coordinates in the table are the average of the corners of the regions. Using these parameters (3.5MW/km$^2$ and 100% utilization rate), the potential capacity of these regions is 192 GW or $12\,802$ turbines. If we instead had used the most optimistic parameters in the report (7.5MW/km$^2$ and 100% utilization rate), the numbers would be 412GW and around $27\,400$ turbines. In the table, we have also included a maximum portfolio weight given a total of 2000 turbines, which is used as a constraint in the portfolio optimization presented in Section 3.

We have also included the average and standard deviation of the hourly capacity factor, estimated from the NORA3-WP data at the closest grid cell to the given coordinates in Table 1. The averages range from 54.6% to 65.6%, and the standard deviations range from 39.2 to 42.4 pp. The expected capacity factor for any portfolio based on these locations will fall in this range of means. We show below that the diversification effect reduces the standard deviation of the total hourly power production considerably.

We refer to these candidate locations as NVE locations.

---

[1]Using 5MW/km$^2$ and 67% utilization would result in 3.35MW/km$^2$ compared to 3.5MW/km$^2$.



| | ID | Location | Longi-tude (°E) | Lati-tude- (°N) | CF Mean (%) | SD (pp) | Area (km²) | Potential installed capacity (MW) | Max NoT | Max portfolio weight |
|---|---|---|---|---|---|---|---|---|---|---|
| N | 1 | Nordavind A | 32 | 71.1 | 57% | 40.3 | 4275 | 14962 | 998 | 49.9% |
| | 2 | Nordavind B | 27.9 | 71.8 | 57.7% | 40.3 | 2239 | 7836 | 522 | 26.1% |
| | 3 | Nordavind C | 20.1 | 71.7 | 56.5% | 40.7 | 1054 | 3689 | 246 | 12.3% |
| | 4 | Nordavind D | 18.7 | 71.4 | 56.3% | 40.8 | 3642 | 12747 | 850 | 42.5% |
| NW | 5 | Nordvest A | 9.5 | 66.2 | 57.6% | 40.9 | 11307 | 39575 | 2638 | 131.9% |
| | 6 | Nordvest B | 7.4 | 64.8 | 56.3% | 41.2 | 3437 | 12030 | 802 | 40.1% |
| | 7 | Nordvest C | 6.8 | 63.8 | 54.6% | 41.8 | 5582 | 19537 | 1302 | 65.1% |
| W | 8 | Vestavind A | 3.7 | 62 | 61.3% | 41.2 | 1884 | 6594 | 440 | 22% |
| | 9 | Vestavind B | 3.8 | 61.1 | 59.3% | 41.8 | 2985 | 10448 | 696 | 34.8% |
| | 10 | Vestavind C | 3.7 | 60.4 | 58.6% | 41.8 | 1040 | 3640 | 243 | 12.1% |
| | 11 | Vestavind D | 4.4 | 60.3 | 55.8% | 42.4 | 724 | 2534 | 169 | 8.4% |
| | 12 | Vestavind E | 3.9 | 59.1 | 61.6% | 40.9 | 1475 | 5162 | 344 | 17.2% |
| | 13 | Vestavind F | 4.5 | 59.2 | 59.8% | 41.3 | 1989 | 6962 | 464 | 23.2% |
| SW | 14 | Sørvest A | 3.5 | 57.9 | 64.1% | 39.9 | 1456 | 5096 | 340 | 17% |
| | 15 | Sørvest B | 3.4 | 57.4 | 64.1% | 39.8 | 2179 | 7626 | 508 | 25.4% |
| | 16 | Sørvest C | 3.9 | 57 | 64.4% | 39.6 | 1766 | 6181 | 412 | 20.6% |
| | 17 | Sørvest D | 3.9 | 56.5 | 63.9% | 39.6 | 1215 | 4252 | 284 | 14.2% |
| | 18 | Sørvest E | 4.7 | 57.5 | 65.5% | 39.5 | 1016 | 3556 | 237 | 11.9% |
| | 19 | Sørvest F | 4.9 | 56.9 | 65.4% | 39.2 | 2702 | 9457 | 630 | 31.5% |
| SE | 20 | Sønnavind A | 7.6 | 57.5 | 65.6% | 39.7 | 2900 | 10150 | 677 | 33.8% |

**Table 1.** NVE selected locations with coordinates and area. The potential installed capacity is calculated using a capacity density of 3.5MW/km² and 100% area utilization rate. The maximum number of turbines (NoT) is based on 15MW/turbine, and CF is the capacity factor. Note that the standard deviation (SD) is measured in percentage points (pp).

## 2.3 Candidate locations based on Solbrekke and Sorteberg (2023)

Solbrekke and Sorteberg (2023) constructed wind power suitability scores (WPSS) for potential wind farm locations for the entire Norwegian Economic Zone (NEZ), taking into account many relevant factors like wind resources, techno-economic as-
pects, social acceptance, environmental considerations and met-ocean constraints such as wind and wave conditions. Solbrekke and Sorteberg (2023) exclude some grid cells due to, e.g. oil platforms or other obstacles, but areas are not excluded solely due to input from one interest group. Since the user must specify the importance of the different criteria, the WPSS is not an objective measure. To cope with the subjective criteria weights, Solbrekke and Sorteberg (2023) carried out a sensitivity analysis where the criteria importance were tuned according to distinct preferences of three actors: *the investor*, *the environ-*
*mentalist*, and *the fisherman*. The result from Solbrekke and Sorteberg (2023) gives information about which NORA3-WP grid



cells are most suited for wind power in NEZ, and the sensitivity analysis reveals which of these are robust to changes in criteria importance.

To avoid some candidates very close to shore, we add a requirement that the offshore location should be at least 15km from the nearest land mass and select locations with WPSS above a certain percentile, $p$, from the baseline scenario and the three

actors of Solbrekke and Sorteberg (2023). To be deemed a suitable location, all three actors and the baseline suitability score must agree that the location is among the top 100p% of candidates, $p \in (0, 1)$. Thus, all candidate locations are grid points with the highest and most robust suitability scores.

Our goal is not to place each wind turbine precisely but, more generally, to place the wind farms. We allow one grid cell ($3 \times 3$km) to represent one wind farm and its surrounding area. Therefore, we do not want to include grid points too close to

each other. If two points are within $r$ km from each other, we select the one with the highest baseline WPSS. The algorithm for doing this is described below, in Algorithm 1. Two choices affect the number of candidate locations: The minimum distance between candidates $r$ and the percentile of WPSS $p$.

---

**Algorithm 1** Algorithm of selecting candidate locations

Let $\mathcal{S}$ denote the set containing candidate locations. A priori, these are all located 15km from shore and among the top $100p\%$ in terms of baseline WPSS and the WPSS of the three actors. Let $r$ denote the minimum distance between candidate locations.

1. Loop over the locations, and for each location $s$, calculate the distance to the other locations. If $\mathcal{R}_s = \{s_* \in \mathcal{S} \backslash \{s\} \colon \|s - s_*\| \leq r\} \neq \emptyset$, select among $\mathcal{R}_s \cup \{s\}$ the one with the highest WPSS and store it in $\mathcal{S}_* \subset \mathcal{S}$.

2. Remove duplicates from $\mathcal{S}_*$ and let $\mathcal{S} = \mathcal{S}_*$.

3. Repeat steps 1-2 until $\mathcal{R}_s = \emptyset$ for all $s \in \mathcal{S}$.

---

We use the top 25% (i.e. p = 0.25) in the Algorithm 1, and the minimum distance between each potential wind farm is set to $r = 40$ km. This is based on the somewhat rough calculation that a wind farm of around 200 turbines will require a

square of $15 \times 15 \, km^2$, and we require at least 10 km between the farms to minimize the interactions between the wind farms. This seems reasonable compared to the minimum distances to the nearest wind farms for existing and planned offshore wind parks (see Figure 3 of Finserås et al. (2024)). The two offshore areas opened for wind farm development in Norwegian waters, UN and SN2, have a planned operational limit of 5km between adjacent wind farms (Norwegian Water Resources and Energy Directorate, 2018). However, the appropriate separation distance between wind farms will greatly vary due to, e.g., atmospheric

stability and wind direction.

The Algorithm 1 is an ad hoc selection procedure to reduce the number of grid points to consider. It does not find a unique and optimal solution to how the Norwegian offshore portfolio should look. This is not a major concern here since the points are only seen as representatives for that area. Running the algorithm with $p = 0.25$ and $r = 40$ km, we end up with 25 locations. For comparison reasons, it is beneficial that the two candidate sets have the same number of locations. Therefore, we increase

the minimum distance between wind farms (r) from 40 km by 1km at a time until 20 locations are selected, including both SN2 and UN.





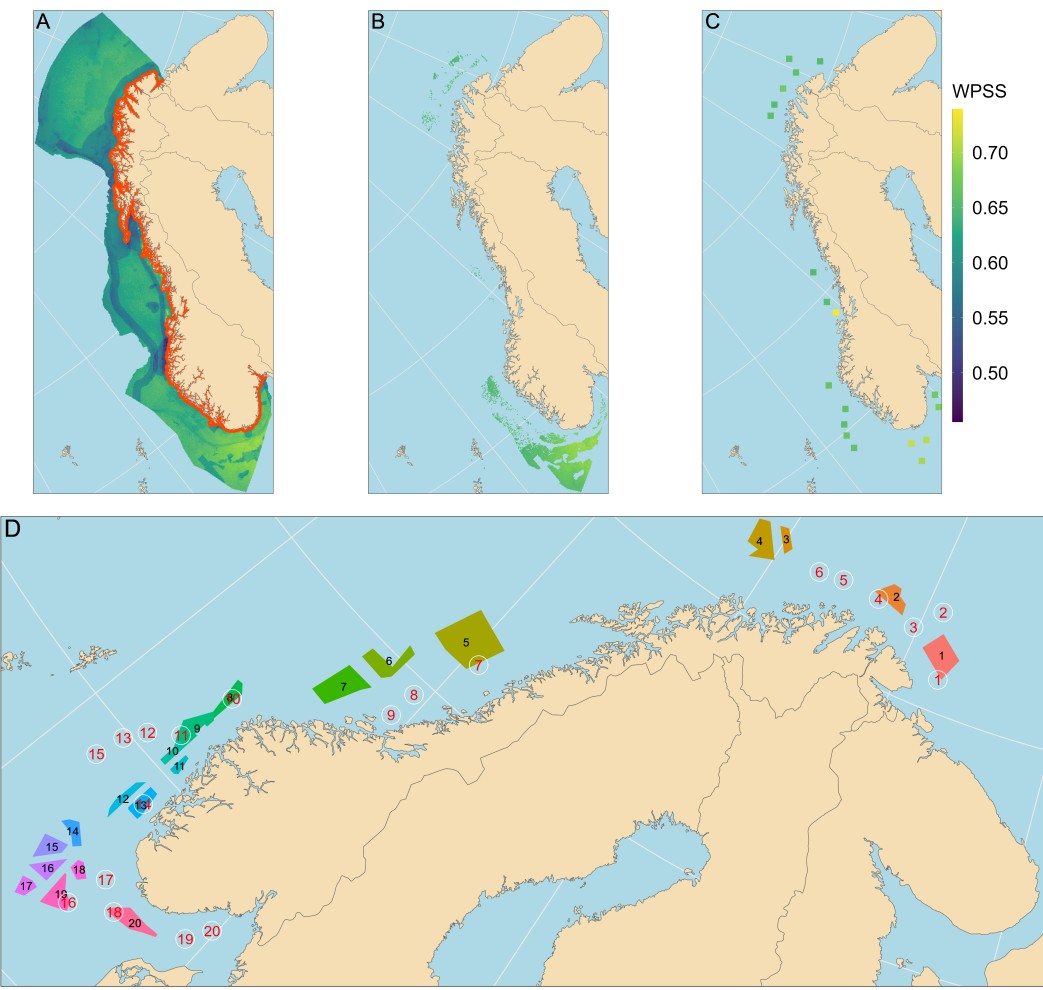

**Figure 1.** A) All locations in NEZ are coloured by wind power baseline suitability score (WPSS). B) Locations that are Top 25% WPSS in all suitability perspectives coloured according to baseline. C) The 19 locations selected after running Algorithm 1. D) The 19 locations chosen as potential wind farm sites with Utsira Nord added, numbered from 1 to 20 in red numbers with white circles around and the 20 NVE-suggested areas numbered from 1 to 20 as shaded areas with black ID numbers (see Table 1).

In Figure 1A, we show all the locations in NEZ considered as potential by Solbrekke and Sorteberg (2023) and being 15 km for shore (total: 71 021 points). In B, we have kept only locations with suitability scores above the 75th percentile in all suitability scores (baseline, investor, fisherman and environmentalist), leaving 6419 suitable locations. Then we run Algorithm 1 with $r = 47$ km and end up with the 19 potential locations in C. For visibility purposes in the figure, we have increased the size of each grid cell by a factor of 10. Among these are SN2 (NVE region Sørvest F) represented, but UN (NVE region Vestavind F) is not. Since the Norwegian Government has decided to allocate areas for wind farms in these two regions, we add UN as a candidate location. This is represented by a grid cell inside NVE region Vestavind F at the centre of the approved



|  | ID | Longi-tude (°E) | Lati-tude (°N) | Mean CF (%) | SD CF (%-pts) | Wind power suitability scores (upper percentile within each actor score) | | | |
|---|---|---|---|---|---|---|---|---|---|
|  |  |  |  |  |  | Baseline | Investor | Fisherman | Environmentalist |
| N | 1 | 32.4 | 70.6 | 56.9% | 40.4 | 9.9% | 12.8% | 18.6% | 14.5% |
|  | 2 | 31.1 | 71.9 | 57.6% | 40.3 | 7.0% | 15.9% | 2.1% | 5.8% |
|  | 3 | 29.7 | 71.4 | 57.1% | 40.3 | 7.4% | 11.5% | 17.1% | 11.0% |
|  | 4 | 27 | 71.6 | 57.7% | 40.3 | 3.9% | 8.0% | 1.8% | 5.6% |
|  | 5 | 24.5 | 71.6 | 55.7% | 40.7 | 9.8% | 14.2% | 24.2% | 23.5% |
|  | 6 | 22.9 | 71.5 | 55.2% | 40.8 | 7.9% | 14.6% | 6.2% | 14.7% |
| NW | 7 | 10.7 | 66 | 56.7% | 41.0 | 10.8% | 19.2% | 6.9% | 16.4% |
|  | 8 | 9.4 | 64.7 | 54.8% | 41.1 | 9.7% | 18.9% | 9.4% | 24.6% |
|  | 9 | 9.2 | 64.1 | 52.3% | 41.0 | <0.1% | <0.1% | 1.7% | 9.9% |
| W | 10 | 3.8 | 62 | 61.2% | 41.2 | 5.8% | 10.9% | 2.1% | 6.0% |
|  | 11 | 3.5 | 60.8 | 59.4% | 41.7 | 6.7% | 11.4% | 3.5% | 8.4% |
|  | 12 | 2.5 | 60.3 | 61.4% | 41.1 | 8.2% | 12.5% | 16.7% | 9.5% |
|  | 13 | 2.1 | 59.9 | 62.3% | 40.8 | 5.6% | 10.9% | 1.6% | 4.9% |
|  | 14 | 4.5 | 59.3 | 59.4% | 41.4 | 14.9% | 11.2% | **45.9%** | **60.9%** |
|  | 15 | 1.9 | 59.2 | 62.9% | 40.5 | 6.8% | 12.8% | 1.7% | 4.9% |
| SW | 16 | 5.3 | 56.9 | 65.7% | 39.1 | 0.3% | 1.5% | <0.1% | 0.1% |
|  | 17 | 5.7 | 57.7 | 66.4% | 39.4 | 0.1% | 0.3% | <0.1% | <0.1% |
|  | 18 | 6.7 | 57.4 | 66.8% | 39.2 | 0.1% | 0.3% | <0.1% | <0.1% |
| SE | 19 | 9.3 | 58 | 60.0% | 40.9 | 2.4% | 2.0% | 1.9% | 3.5% |
|  | 20 | 9.8 | 58.5 | 56.3% | 41.3 | 3.9% | 3.2% | 6.9% | 13.2% |

**Table 2.** ID, location, mean and standard deviation of capacity factor (CF) for the S&S selected candidate locations. The latter four columns are the wind power suitability scores of the different actors presented by which percentile of the score in NEZ by Solbrekke and Sorteberg (2023). Except for location 10, all are below 25% by assumption, with distance to shore above 15km.

region identified by Norwegian Water Resources and Energy Directorate (2012). This point is more than 40 km away from any other location. It has a baseline suitability score of 14.85% and investor score of 11.2%, but fisherman and environmentalist of respectively 46% and 60.9% which violates the top 25% assumption (see Table 2). Thus, the 20 locations are shown in Figure 1D, numbered from 1 to 20 following the Norwegian coast from the Barents Sea to Skagerak, where location 14 is in UN and location 16 is in SN2. We will refer to this set of candidate locations as S&S. There is some overlap between the candidate sets (Figure 1D). Some S&S locations lie within a corresponding NVE region (S&S10-Vestavind A, S&S11-Vestavind B, S&S18-Sønnavind A) or just outside (S&S1-Nordavind A, S&S4-Nordavind B, S&S7-Nordvest A).

In Table 2, we have presented the coordinates of the selected candidate locations with the mean and standard deviation of the capacity factor based on the NORA3-WP grid cell. The suitability scores are calculated on the same grid as NORA3-WP,





so the coordinates are exact (although rounded off). We have also included the wind power suitability scores as what upper percentile the location has within each actor score. Some of the scores of location 14 (UN) are highlighted as they violate the

top 25% score assumption for the fisherman and environmentalist. The mean capacity factor ranges from 52.3% to 66.8%, while the standard deviations range from 39.1 to 41.7 pp. These locations thus have a wider range of mean capacity factors and a narrower standard deviation range compared to the NVE regions. S&S location 18 has the highest mean capacity factor of 66.8% and the second lowest standard deviation of 39.2 pp, making it probably the best location among all the candidates. It resides within the NVE region Sønnavind A, which also has the highest mean capacity factor of the NVE regions (Table 1).

We divide the two candidate location sets into five regional groups, corresponding to the Norwegian naming of the NVE regions: North (N), North-West (NW), West (W), South-West (SW) and South-East (SE). Which locations belong to which group can be seen from the first columns of tables 1-2. The northern groups are similar across NVE and S&S, but the NVE regions are more spread out, while the S&S is closer to shore (except S&S2). We have the same number of locations and roughly the same geographical spread in the North-West. The largest difference is that the S&S locations are much closer to

shore, especially S&S9. The opposite is true for the west coast, where S&S12-13 and 15 are further out at sea than the NVE regions. The other S&S locations overlap with NVE regions for this group. The most striking aspect in the South-West group is that there are twice as many NVE regions as there are S&S regions. The NVE regions are also farther to the south and west. S&S18 lies inside Sønnavind A, but we count S&S18 as South-West, while Sønnavind A is the only NVE member in the South-East. Farther east, we also find S&S19-20 in this group, closer to the Danish-Swedish-Norwegian border intersection

in the Skagerak Sea. The two candidate sets both have some similarities and some interesting distinctions. This grouping is applicable when interpreting the correlation structure of the two location sets (see section 3.1).

The NVE regions have a finite area, which we use for setting constraints on the maximum number of turbines. The S&S areas do not have this. To ensure a fair comparison, we do not allow S&S locations with more than 500 turbines. The median maximum number of turbines for the NVE regions is 486, so we have rounded this off to 500. This constraint corresponds to

an area of 2143 km$^2$ using the same assumptions of 100% utilization and 3.5MW/km$^2$, comparable in size to Sørvest B or Nordavind B.

Now that we have our two sets of candidate locations, NVE- and S&S locations, we present a methodology for allocating turbines to the different wind farm locations.

## 3   Modern portfolio theory

There are some fundamental differences between a wind farm portfolio and a portfolio of financial assets:

1. One cannot borrow turbines (which corresponds to shorting assets in finance), meaning the portfolio weights have to be non-negative.

2. Selling or re-balancing the portfolio as time passes is not feasible. One can, in practice, only build more wind parks by investing more.





3. In financial investment theory, a higher risk should give a higher potential return, which is not necessarily valid for wind power production.

4. There is no equivalent to a risk-free interest rate for a portfolio of wind farms.

We argue, however, that the well-known diversification effect that we get when spreading financial investments across a large portfolio of assets is highly relevant when selecting sites for wind power production as well. In finance, we look for assets that have low, or even negative, correlation to lessen the impact of negative movements in the markets on the overall portfolio. If the value of one investment goes down, this will not be systematically associated with failure in other investments simultaneously. The corresponding phenomenon for wind farms is if wind conditions at one wind farm location are systematically associated with conditions at other locations. As one increases the distance between wind farms up to a point, the systematic association decreases, and the diversification effect increases (Solbrekke et al., 2020; St.Martin et al., 2015).

The classical approach for allocating wind farms seen in previous studies is modern, or mean-variance, portfolio theory (Markowitz, 1952). The goal is, for a given target capacity factor, $T_{\mathrm{CF}}$, to compose the portfolio that exhibits the minimum variance. Let $X_{ti} \in [0,1]$ denote the stochastic variable for capacity factor from location $i$ at time $t$ and $\mathbf{X}_t = (X_{t1},\ldots,X_{tm})'$, where $m$ is the number of locations, $t = 1,\ldots,n$ and $'$ is the transpose operator. Let $\boldsymbol{\mu} = (\mu_1,\ldots,\mu_m)' = \mathbb{E}\mathbf{X}_t'$ denote the time invariant expected value vector. Further, let $\boldsymbol{\Sigma} = \mathrm{Cov}(\mathbf{X}_t)$ denote the covariance matrix. Let $\mathbf{w} = (w_1,\ldots,w_m)'$ denote the vector of non-negative portfolio weights, i.e. the proportion of the total number of wind turbines installed at each location. These must be non-negative because you cannot build a negative number of wind turbines. The portfolio capacity factor at time $t$, $Y_t$, can then be expressed as

$$Y_t = \sum_{i=1}^{m} w_i X_{ti} = \mathbf{w}' \mathbf{X}_t.$$

The expected (time-invariant) capacity factor and the corresponding variance of the wind farm portfolio are respectively given by

$$\mathbb{E}Y_t = \sum_{i=1}^{m} w_i \mu_i = \mathbf{w}' \boldsymbol{\mu} \quad \text{and} \quad \mathrm{Var}(Y_t) = \mathbf{w}' \boldsymbol{\Sigma} \mathbf{w}.$$

We want to find a portfolio that minimizes the portfolio variance for a given level of expected capacity factor; $T_{\mathrm{CF}}$. We require the weights, $\mathbf{w}$, to sum to one, i.e. $\mathbf{w}'\mathbf{1} = 1$, where $\mathbf{1} = (1,\ldots,1)'$ is a vector of length $m$. Then, we can formulate the optimization problem as

Minimize:  $\mathbf{w}' \boldsymbol{\Sigma} \mathbf{w}$,

Subject to:  $\mathbf{w}' \boldsymbol{\mu} = T_{\mathrm{CF}}$  $\mathbf{w}'\mathbf{1} = 1$,  and  $w_i \geq 0, i = 1,\ldots,m.$

This is the simplest case with the bare minimum of constraints and corresponds to what Markowitz (1952) also used.

The formulation above does not exclude solutions where, for instance, only one wind turbine is placed far out in the Barents Sea or some turbines at every location. Small, spread-out wind farms are not a realistic solution since one must invest a lot in the infrastructure associated with each farm. Therefore, we rather prefer to cluster together many turbines at a few locations, which can be achieved by adding a constraint on the maximum number of nonzero weights, i.e. locations with more than





zero turbines, called a *position limit constraint*. Another relevant constraint is the so-called *box constraint*, meaning setting a
lower and upper limit on each location's weights and thus restricting the number of turbines allowed at said location. The box
constraint avoids too large or too small wind farms. We derive maximum constraints based on the limited areas of the NVE
regions (Table 1) and use a maximum of 500 turbines per location for the S&S locations. Tejeda et al. (2018) also use a box
constraint with minimum 0 and maximum 250MW per grid cell ($\approx 550$ km$^2$).

For the position limit constraint, let $1(w > 0)$ denote the indicator function, which equals 1 if $w > 0$ and 0 otherwise and
let $h$ denote the maximum number of nonzero turbine locations. For the box constraint, let $w_i \in [\ell, u]$ for $i = 1, \ldots, m$, where
$0 \le \ell \le u \le 1$. We write $w_i \in [\ell, u] \cup \{0\}$ to allow for zero weights if combined with a position limit constraint.

The problem then becomes

Minimize:    $\mathbf{w}' \mathbf{\Sigma} \mathbf{w}$,        (Objective)

Subject to:    $\mathbf{w}' \boldsymbol{\mu} = T_{\text{CF}}$,        (Expectation constraint)

$\mathbf{w}' \mathbf{1} = 1$,        (Sum-to-one constraint)

            $w_i \in [\ell, u] \cup \{0\}, i = 1, \ldots, m$,        (Box constraint)

and    $\sum_{i=1}^{m} 1(w_i > 0) \le h$.        (Position limit constraint)

We consider five different scenarios with different combinations of constraints below. The most optimal solution would be the
one with the fewest constraints, but it may be unrealistic due to the reasons listed earlier in this section.

We estimate the expected value vector $\boldsymbol{\mu}$ and the covariance matrix $\mathbf{\Sigma}$ by the empirical mean and covariance of the hourly
observations, i.e.

$$\widehat{\boldsymbol{\mu}} = \frac{1}{n} \sum_{t=1}^{n} \mathbf{X}_t, \quad \widehat{\mathbf{\Sigma}} = \frac{1}{n-1} \sum_{t=1}^{n} (\mathbf{X}_t - \widehat{\boldsymbol{\mu}})(\mathbf{X}_t - \widehat{\boldsymbol{\mu}})'.$$

The target capacity factor, $T_{\text{CF}}$, should be within the range of $\widehat{\boldsymbol{\mu}}$. Otherwise, a solution will not exist. If $T_{\text{CF}} = \max_i \widehat{\mu}_i$ all
turbines must be placed on the location with the highest mean capacity factor.

We implement the position limit constraint by optimizing all combinations of $h$ locations separately and selecting the mini-
mum variance portfolio. We compare this approach to a step-wise approach by adding the location that improves the portfolio
performance the most in each step, referred to as a sequential build-out. In the sequential build-out, we start building on the two
locations the Norwegian government already have decided on (1500 MW at each) and then consider adding one other location
(looping over all the other candidates) or building more turbines at the existing locations. We use the lower box constraint to
keep the already-built turbines in the next iteration.

The number of turbines at a location is an integer number. In the portfolio optimization, we estimate weights as the proportion
of 2000 turbines (30 GW). We get the number of turbines at a location by multiplying 2000 turbines with their portfolio weight
and rounding off to the nearest integer. Even though the sum of the weights is one, this rounding off may lead the total number
of turbines to not equate 2000. We compensate for this by removing one turbine from the necessary number of locations in
the estimated portfolios with too many turbines until the total is 2000 and correspondingly adding one turbine to the necessary





number of locations with too few. The identified locations where this adjustment is applied are those with the number of turbines closest to being rounded down or up, respectively.

We use the R package *quadprog* (Turlach et al., 2019), which contains functions for solving quadratic programming problems, to optimize the portfolios under different constraint scenarios. For reproducibility purposes, the R code is made available at https://github.com/holleland/OffshoreWindPortfolios (see data availability statement for further details).

### 3.1 Correlation structure

From the NORA3-WP dataset, we estimate a mean vector $\widehat{\boldsymbol{\mu}}$ and an empirical covariance matrix $\widehat{\boldsymbol{\Sigma}}$ for the NVE regions and the corresponding S&S locations. The covariance structure is important for the diversification effect. To simplify, say we have a portfolio of two assets, X and Y, such that the portfolio value is $wX + (1-w)Y$, with weight $w \in (0,1)$. The portfolio variance is then

$$w^2 \sigma_X^2 + (1-w)^2 \sigma_Y^2 + 2w(1-w)\rho\,\sigma_x\,\sigma_y,$$

where $\rho \in [-1,1]$ is the correlation between X and Y, $\sigma_X$ and $\sigma_Y$ are the standard deviations of X and Y, respectively. All else

fixed, reducing the correlation $\rho$ will thus reduce the portfolio variance. Having wind power locations with a low correlation with other locations (perhaps even negative) is important for achieving diversification effects and a lower portfolio standard deviation.

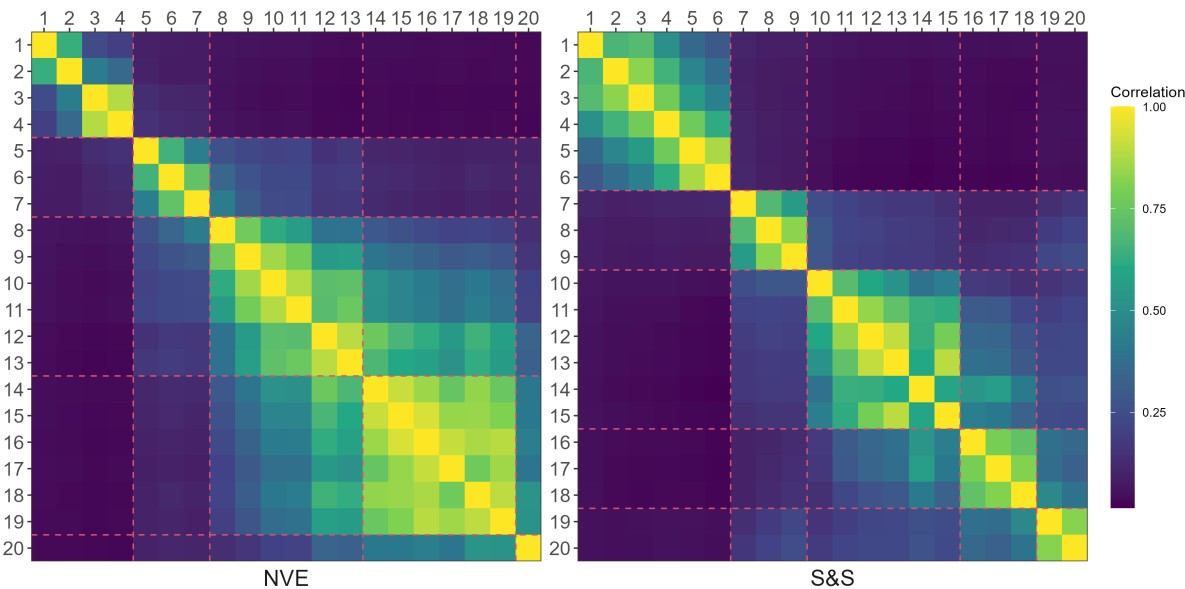

**Figure 2.** Correlation matrices for the hourly capacity factor at the 20 NVE regions and S&S locations. The red dashed lines split the locations according to the grouping of tables 1-2.





We have presented the correlation matrices for NVE- and S&S locations as correlation heat maps in Figure 2. Since the standard deviations are all in the same range (around 40pp, see tables 1 and 2), we use correlations instead of covariances as the scale is easy to interpret. The red dashed lines split the correlation matrices into blocks corresponding to the five location groups N, NW, W, SW and SE, described above. We have a substantial block structure following this grouping in the matrices, where the correlations are high within each block but low between them. The locations farthest to the north (Nordavind A-D, S&S1-6) are almost uncorrelated with the remaining locations. The clear block off the northwest coast (Nordvest A-C and S&S7-9) also stands out as having a low correlation with the other blocks.

Further south, more locations are more closely packed, leading to a higher correlation between the blocks. The between-blocks correlations might be slightly higher for the southern NVE regions than the S&S locations, but these regions are also more evenly spread. In contrast, the S&S locations are more clustered in three more distinct groups (see map in Figure 1D). For the NVE regions, Sønnavind A stands out as a location with low correlation with others, and similarly for the two S&S locations in Skagerak (S&S19-20).

## 3.2 Portfolio cases

We set up five sets of constraints under which we find the optimal portfolio. For scenarios below, with exactly five locations, our greedy algorithm for the position limit constraint loops over all combinations of 5 locations. For cases B and C, there are $\binom{18}{3} = 816$ and $\binom{20}{5} = 15\,504$ such combinations, respectively, using the binomial coefficient notation. The cases are as follows:

**Case A** No constraints on the number of locations (benchmark portfolio).

**Case B** Exactly 5 locations, where Sørlige Nordsjø 2 (SN2) and Utsira Nord (UN) are included.

**Case C** Exactly 5 locations, not necessarily including SN2 and UN.

**Case D** Having built 1500 MW at SN2 and 1500 MW at UN. Where to build 1500 MW next? ("as we go" / sequential build out) until reaching 30 GW total. At each step, we either build where we already have farms or add one more location.

**Case E** Only building on the first 5 sequentially selected locations from Case D.

The resulting portfolios will depend on the value used for the target capacity factor, $T_{\mathrm{CF}}$. For comparison purposes, $T_{\mathrm{CF}}$ should be within the range of values for both the NVE regions and S&S locations. We therefore run the portfolio optimization for three values of $T_{\mathrm{CF}}$ being 58%, 60% and 62%. For case A, we also solve the problem with a high resolution of $T_{\mathrm{CF}}$ between 56% and 65%. These are high capacity factors compared to onshore wind farm portfolios, e.g. Tejeda et al. (2018) use 23% capacity factor for their mainly onshore setting.

Case A will not necessarily give a realistic portfolio of wind farms as it has no restrictions on the number of locations. The optimal solution may involve many locations, some very small. However, it should give the portfolio with the lowest standard deviation and, as such, it is a meaningful benchmark for the other portfolios. For cases B, C, and E, we are restricted to only





building on five locations, which will not result in very small wind farms. An interesting comparison is between Case B and E
as both require building on SN2 and UN and three other locations. They may result in the exact same portfolios. For the NVE
regions, we do not allow for more turbines than the maximum number of turbines given in Table 1, and correspondingly, not
more than 500 turbines for S&S locations. Note that for case E, we initially build five wind farms as in Case D, corresponding
to 7.5 GW installed capacity, and then distribute the remaining 22.5 GW on these five locations.

The sequential build-out in cases D and E starts with a portfolio of 1500 MW (100 turbines) at NVE regions Sørvest F and
Vestavind F and, correspondingly, S&S locations 16 and 14. These are the locations where the Norwegian Government has
decided to start building the first offshore wind farms, i.e. SN2 and UN. The initial portfolio of 1500 MW at each location will
not fulfil the $T_{CF}$ requirement, as the capacity factor will be 62.6%. We then, for all combinations of the existing wind farms
and one new candidate location, find the optimal portfolios, using a box constraint to ensure that at least 1500 MW is kept on
SN2 and UN. We also consider not adding new locations but merely building more on existing ones. Having found the optimal
portfolios for each candidate location, we choose the one that minimizes the portfolio standard deviation and fulfils the target
capacity factor constraint. We then update the box constraint with the weights of the best portfolio, and we start a new iteration,
considering adding another location or building more on the existing ones in the same manner. In each iteration, we build 1500
MW (100 turbines) and repeat the process until 30GW installed capacity or 2000 turbines have been placed. All the portfolios
following the initial one must fulfil the $T_{CF}$ requirement. Since SN2 and UN have such a high mean capacity factor, we do not
run cases D and E for $T_{CF} = 58\%$. Having initially installed 1500 MW at SN2 and UN, and then deciding where to place the
next 1500 MW to achieve 58% capacity factor, the only choice would be to place all 1500 MW at a location with a capacity
factor around 49%. The minimum capacity factor for the NVE- and S&S locations, respectively, is 54.6% and 52.3%. Hence,
there exists no solution for the first iteration of the sequential algorithm when the target capacity factor is 58%.

### 3.3 Underlying assumptions

For the analysis that follows, we make some simplifying assumptions that we want to make explicit. We refrain from imposing
the current limitations on the Norwegian transmission grid to the optimization problem presented in this paper because they
will likely change over the decades to come. Instead, we assume a "copper-plate Norway", where the power grid is fully
connected so that power production in the Barents Sea can have beneficial diversifying effects on the total power production in
the event of no wind in the North Sea, for instance. Some offshore wind farms may not even produce energy for the Norwegian
energy system but exclusively export energy to other European countries. The copper-plate assumption simplifies the problem
but is not essential to the methodological approach that we propose. One alternative could be to assume no, or a very limited,
transmission between the North and the South of Norway, which is more realistic today. This would then split the problem into
two separate parts, on which we can apply the same analytical strategy separately, where the government distributes the total
amount of installed power, 30 GW, say, between the two regions.

The wind power generated from the offshore fleet will enter an existing power market and become a portion of the higher-
level electric power portfolio. One could imagine optimizing the allocation of wind power for this portfolio considering the
current power sources. Hydropower is the dominating energy source for producing electricity in Norway today (88.2% in





2022, Statistics Norway (2023)). While hydropower, in most cases, can be controlled by opening or closing the flow of water, wind power is a non-dispatchable energy source. It must be utilized instantly unless stored (e.g., charging batteries, producing

hydrogen or pumped hydro storage). Therefore, it is appropriate that Norwegian hydropower will adapt to wind power rather than vice versa. Even though the combination of wind power and hydropower is an interesting case study, we focus exclusively on the wind power portfolio.

Primary drivers of offshore spatial planning will likely be investment- and maintenance costs. Costs will undoubtedly vary across different sites. For the S&S locations, factors that affect costs are implicitly regarded through the suitability score of each

330 region. The suitability scores consider cost-increasing factors such as ocean depth and distance to shore, which are especially important for the investor actor's suitability score (Solbrekke and Sorteberg, 2023). After selecting the candidate locations, we only optimize the portfolios based on wind power resources and do not consider costs. A key parameter for a cost-benefit analysis is the price of electricity, which is difficult to forecast far into the future. Historical prices are irrelevant since 30 GW of wind power will nearly double the Norwegian electricity production (not considering changes in supply from other sources).

A further analysis considering prices and costs may be a way to take this further, but it is outside the scope of this paper.

## 4   Results

In Figure 3 we have plotted the optimal portfolios, with portfolio standard deviation on the x-axis and portfolio capacity factor on the y-axis, where the colour distinguishes the two sets of locations we consider. In addition to the cases, we have included

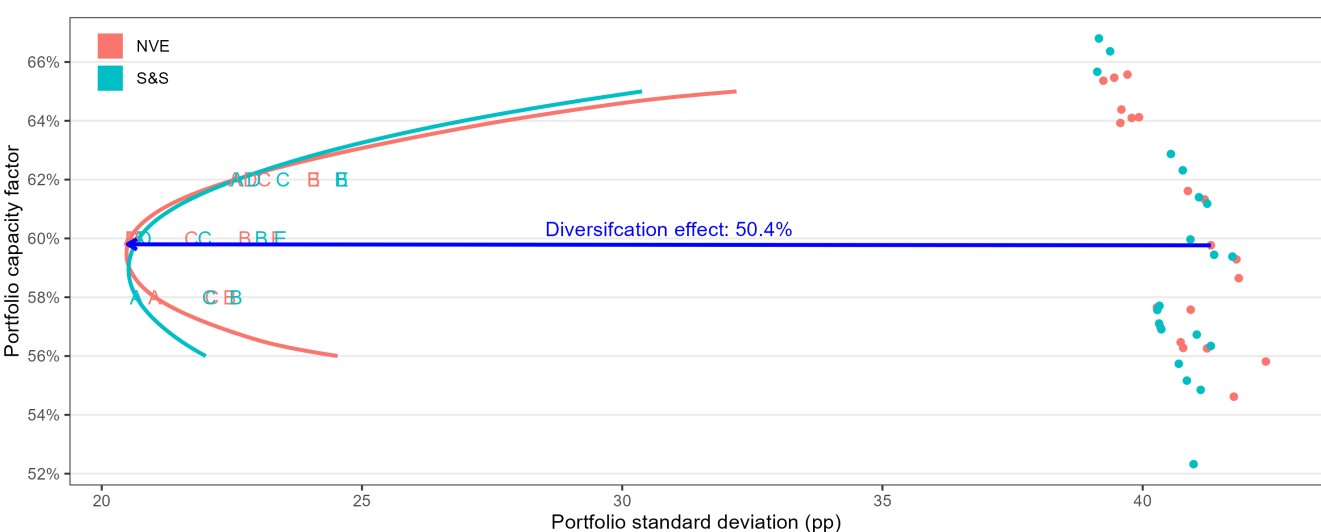

**Figure 3.** Portfolios summarized by portfolio capacity factor and standard deviation in percentage points (pp) for NVE and S&S locations. The dots are one location portfolios, while the letters represent the different case portfolios for capacity factor targets 58%, 60% and 62%. The curved lines are efficient frontiers for case A, i.e. optimal portfolios for a finer sequence of capacity factors from 56 to 65%.





*single-location portfolios* as coloured dots, i.e. portfolios with all turbines placed at one location, even if this would violate

the size constraints. These dots are simply the hourly mean and standard deviation of the capacity factor at each location. The

|  | $T_{CF}$ | Case | SD (pp) | NoWF | Min Turb. | Max Turb. | CF 5% | CF 95% |
|---|---|---|---|---|---|---|---|---|
| NVE | 58% | A | 21.0 | 12 | 20 | 333 | 22% | 91.4% |
|  |  | B | 22.5 | 5 | 300 | 479 | 19.6% | 95.5% |
|  |  | C | 22.1 | 5 | 283 | 468 | 20.5% | 94.6% |
|  | 60% | A | 20.5 | 15 | 25 | 333 | 24.3% | 91.9% |
|  |  | B | 22.8 | 5 | 158 | 495 | 20.3% | 98.5% |
|  |  | C | 21.7 | 5 | 309 | 526 | 22.3% | 95.2% |
|  |  | D | 20.6 | 14 | 11 | 321 | 24.2% | 91.9% |
|  |  | E | 23.4 | 5 | 279 | 522 | 19.3% | 100% |
|  | 62% | A | 22.6 | 11 | 43 | 510 | 22.2% | 94.8% |
|  |  | B | 24.1 | 5 | 183 | 523 | 19.7% | 100% |
|  |  | C | 23.1 | 5 | 276 | 525 | 21.2% | 98% |
|  |  | D | 22.9 | 11 | 63 | 527 | 21.7% | 95% |
|  |  | E | 24.1 | 5 | 183 | 523 | 19.7% | 100% |
| S&S | 58% | A | 20.7 | 17 | 4 | 311 | 22.7% | 91.1% |
|  |  | B | 22.6 | 5 | 268 | 471 | 18.6% | 95.3% |
|  |  | C | 22.1 | 5 | 348 | 434 | 20.7% | 94.3% |
|  | 60% | A | 20.7 | 15 | 14 | 246 | 24% | 92.2% |
|  |  | B | 23.1 | 5 | 179 | 500 | 19.6% | 99% |
|  |  | C | 22.0 | 5 | 316 | 500 | 21.8% | 95.7% |
|  |  | D | 20.8 | 13 | 50 | 222 | 23.8% | 92.5% |
|  |  | E | 23.4 | 5 | 191 | 500 | 19.1% | 99.9% |
|  | 62% | A | 22.6 | 11 | 59 | 451 | 21.9% | 95.4% |
|  |  | B | 24.6 | 5 | 218 | 500 | 18.2% | 100% |
|  |  | C | 23.5 | 5 | 265 | 500 | 20.6% | 100% |
|  |  | D | 22.9 | 12 | 43 | 495 | 21.3% | 95.6% |
|  |  | E | 24.6 | 5 | 262 | 500 | 18.3% | 100% |

**Table 3.** Summary of the different wind farm portfolios for the two sets of potential locations, cases and expected capacity factor (CF) with corresponding power output in GW in parenthesis. Correspondingly, the standard deviation (SD) of the portfolio is given both in percentage points (pp) and GW. The number of wind farms (NoWF) with turbines, and the smallest (Min Turb.) and largest (Max Turb.) The CF 5% and 95% columns are the 5- and 95-percentiles of the portfolio capacity factor.





curves are for case A with values of the capacity factor target between 56 and 65%. Since case A has the mildest restrictions, the curves represent the minimum standard deviation possible for this range of target values. These curves are called the *efficient frontiers* in modern portfolio theory. We note that the size of the diversification effect is large. Distributing wind turbines across multiple sites cuts the portfolio standard deviation nearly in two, from approximately 40% to approximately

20%, compared with collecting all turbines in a single location. The minimum standard deviation portfolios based on NVE and S&S have standard deviations of 20.5 pp and expected capacity factors of 59.5% and 58.9%, respectively. Vestavind F has a mean capacity factor of 59.8%, which is quite close to 59.5% of the NVE minimum standard deviation portfolio and a corresponding standard deviation of 41.3pp. Compared to the 36% reduction found by Drake and Hubacek (2007), we find a potential geographical diversification effect of 50.4% from placing all power at Vestavind F to the minimum standard deviation

portfolio of the NVE locations at $T_{CF} = 59.8\%$, indicated by the blue arrow in Figure 3.

For financial investments, intuition says a higher risk should give a higher potential return. In our case of wind power production, however, we see that the highest power-producing locations also have the lowest standard deviations. The efficient frontier curves in Figure 3 illustrate this point on a portfolio level. As the target capacity factor decreases, the standard deviation decreases to a point (around $T_{CF}$=59% ), where it turns. Decreasing the target capacity factors beyond this point will increase the

portfolio standard deviation. Therefore, the portfolios for case A at 58% have a higher standard deviation than the corresponding at 60% and should never be chosen. The same effect is present in case C going from 58 to 60%. The diversification effect on the portfolio standard deviation is strong, so the best portfolios must include some relatively high- and low-producing locations. However, to achieve a 58% capacity factor, the portfolio must consist of more low mean locations with higher standard deviation. The reason for this U-turn is the lack of a risk-free asset among the candidate locations. There is no

equivalent to placing capital *in the bank*, so to speak, at a risk-free interest rate in wind power production, which is necessary for a monotonically increasing efficient frontier.

Case A has fewer constraints than any other scenario and should give the portfolio with the lowest standard deviation for the same capacity factor target. We can see from Table 3, presenting summary statistics for the different portfolio cases, and Figure 3 that this is the case. Without any constraints on the number of locations, case A tends to have many wind farms,

ranging from 11 to 17 across all capacity factor targets and location sets. The Table A1 shows the actual number of turbines per wind farm location, and the turbines for case A are spread out across NEZ, although less for the 62% target capacity factor cases. It is relevant to compare case A to the sequential build-out case D, as both have no restrictions on the number of wind farms. The number of wind farms for these cases is similar. However, we have small wind farms ranging from 4 to 63 turbines. Restricting the optimization problem to 5 locations (cases B, C and E) seems to avoid this issue because they result in the

minimum number of 158 turbines at one location. We impose restrictions on the maximum number of turbines for the different locations. The largest NVE farm is 527, while, for S&S cases, the 500-turbine upper limit is met in six of the thirteen scenarios.

For cases B, C, and E, we build five wind farms, and in cases B and E, UN and SN2 are required to be included. In terms of standard deviation, the ascending order of these cases should always be C-B-E. This is because case C has the least requirements. It does not need to include UN or SN2. Case B has to include UN and SN2, but we can build as few turbines

as we want there. For case E, we must have at least 1500 MW (100 turbines) installed capacity on both UN and SN2, as




this is the initial portfolio. From Table 3 and Figure 3, we see that the ascending order holds, but for capacity factor 62%, cases B and E have the same standard deviation up to the accuracy of the table. Looking at the exact distribution of the 2000 turbines, presented in appendix Table A1, we see that the portfolios are exactly the same for the NVE locations but with minor differences for the S&S locations.

For the sequential build-out in case D, we have plotted the decreasing portfolio standard deviation as a function of the sequentially increasing installed capacity in Figure 4. We have included the case A standard deviation for the respective setups as dashed horizontal lines, representing the lower threshold of what is possible to achieve. The numbers correspond to the number of wind farms included in the portfolio at the given iteration. The diversification effect is apparent. As expected, the portfolio standard deviation decreases rapidly as the number of locations increases, converging towards an asymptote. Building

around 5-7 wind farms achieves a high diversifying effect on the standard deviation. In fact, with only 3 locations for the NVE case with a target capacity factor of 60%, the reduction in standard deviation compared to placing all turbines at Vestavind F is roughly the same as what Drake and Hubacek (2007) found (37.8%). This estimate is conservative as the target capacity factor is lower at Vestavind F compared to the 60% target. A supplementary animation for case D at 60% target capacity factor for the NVE regions, showing the turbine distribution on a map for each iteration, is available at the GitHub repository associated

with the article (see the data availability statement). Remember that the initial portfolios with only two locations (UN and SN2) are not optimized, i.e. the weights are not estimated but fixed to 0.5 and 0.5. Therefore, the initial portfolio does not fulfil the target capacity factor constraint. For the 62% cases, we see that after having installed 16.5GW on 11 locations and 18GW on

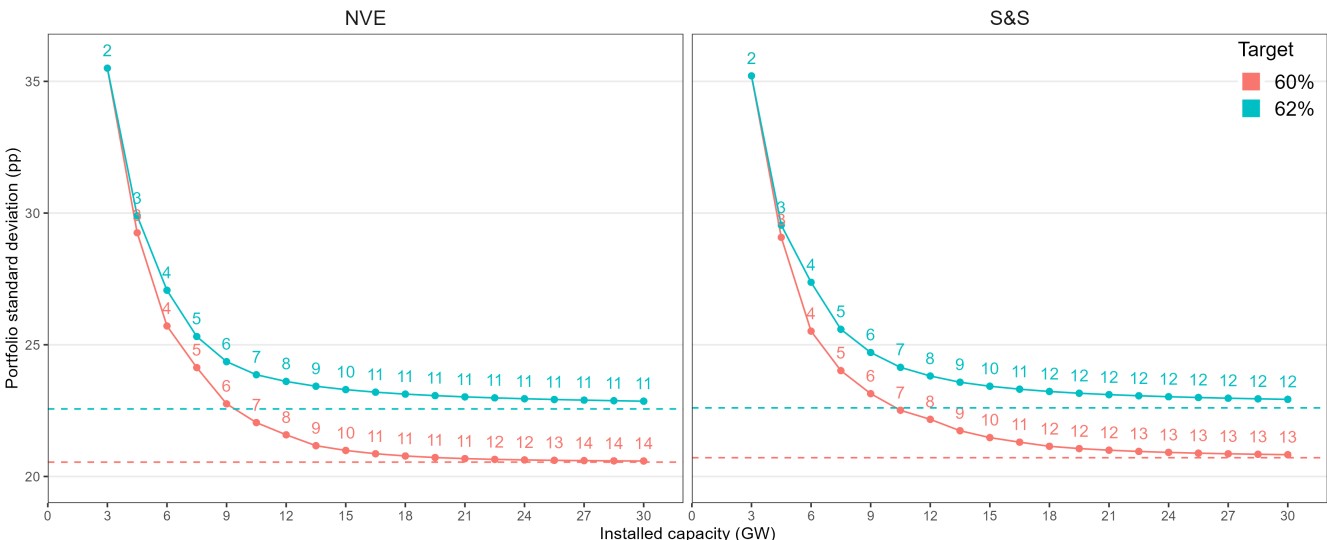

**Figure 4.** Standard deviation (pp) as a function of installed capacity (GW) for sequential build-out in case D, coloured by target capacity factor and panels by candidate location set. The numbers give the number of wind farms at each iteration. The dashed lines are the standard deviations for the corresponding case A portfolios.





12 locations, we do not include new NVE- and S&S locations, respectively. It is better to extend the existing ones. At 60%, the corresponding numbers are 27GW at 14 and 22.5GW at 13 locations. We can also see from the figure that the portfolio standard deviation is approaching the case A value with a negligible difference, and for 60% NVE, it reaches it exactly. Note that case E does not have the same standard deviation as the five wind farms point in Figure 4, because at that point in the build-out, UN and SN2 have minimum 20% of the installed turbines each, while, for case E at 30 GW, they are only required to have 5%.

We see in Figure 3 that the diversification effect of 50.4% would have been nearly the same if we compared against the S&S case A efficient frontier. Having the same number of locations and similar box constraints for the wind farm locations, we can also compare the portfolios between the two sets of locations. From Figure 3 and Table 3 the NVE portfolios outperform the S&S portfolios case by case in terms of variance for the most relevant capacity factors 60% and 62%, but the differences are slight. The minimum standard deviation on the efficient frontiers is the same, but the NVE has a higher capacity factor at that point on the curve. For capacity factors below this minimum standard deviation portfolio, the S&S locations have lower standard deviations for case A and almost the same for cases B-C. The only notable case in terms of robustness is case B as it has a lower standard deviation at 58% than at 60%. Above 62%, the efficient frontier of S&S lies above NVE with a large difference at 65%. This is because, at 65%, we approach the maximum capacity factor of the NVE regions (65.6%). At the same point, S&S still has three locations well above 65% and thus has a higher potential for diversification.

As the Norwegian Government has decided to start building wind farms at SN2 and then UN, they are currently on a sequential build-out strategy. It is interesting to compare this to a global optimization scenario where the perspective is how the offshore wind power portfolio looks when all is said and done, i.e. when all the offshore areas for the 30GW wind power are settled. We have already seen that the standard deviation of the sequential case D approaches that of A, but the distribution of the turbines is also essential. Comparing case A to case D in Table A1, they seem very similar in terms of where to build the wind farms. In most cases, where a different location is selected, it is a neighbouring location.

We note that UN is not included in case A for portfolios with a capacity factor above 58% (Table A1). At 62%, cases A and D have selected the same locations except for UN (Vestavind F / S&S14). Interestingly, in the iterations of the sequential build-out, we never build any turbines at UN except for the 100 we started with (Tables A2-A3 in appendix). In fact, in all the cases where UN is not required (A and C), we do not build any turbines there if $T_{CF} > 58\%$, suggesting that UN is likely not a highly suitable location when only concerned with minimizing the standard deviation and under our other assumptions. On the other hand, SN2 is included in all S&S A cases and NVE cases except the 58% capacity factor. From Table A1, we can also see that SN2 is included in all cases when $T_{CF} = 62\%$.

For the other locations, Sønnavind A/S&S18 and Vestavind A/S&S10 stand out as locations selected in most cases. These location pairs are almost the same in both sets. Sønnavind A/S&S18 have the highest capacity factor and among the lowest standard deviations, making it an attractive location. Vestavind A/S&S10 is likely included due to its diversification effect. Among the S&S locations, S&S10 is far from the central cluster off the West Coast and the furthest away from UN for cases where this is mandatory. The latter also holds for the NVE region. Based on the correlation matrices and with a maximum of five wind farms, choosing one location from each of the five regional groups seemed natural to obtain the most considerable





diversification effect. For cases B, C, and E, such a clear pattern is not seen. We always place one or two wind farms in the North- and Western groups. If two wind farms, they are always the ones that are farthest away from each other (Nordavind A and D, Vestavind A and F, S&S1 and S&S6, S&S10 and S&S14). In the South, Sørvest F/S&S16, Sønnavind A/S&S18, or both are included. There is a tendency to follow the correlation matrix blocks (see Figure 2) for the 60% case C, where we get two farms in the North, one in the North-West, one in the West and one in the South. At 62%, for the same case, turbines are placed at the two farms in the South probably to achieve the capacity factor requirement.

One interesting point is that for the 60% capacity factor case D, the first selected location S&S8 and Nordvest C, all and almost all of the 100 new turbines placed in that iteration are put at the new site (see Tables A2-A3), respectively. Choosing these locations and placing virtually all new turbines there pulls the capacity factor from 62.6% to 60%. We do not build any more turbines at S&S8; for Nordvest C, only three turbines follow the initial 98. This observation indicates that these selected locations in the first iteration are sub-optimal for minimizing the variance, as the procedure has to assign turbines to them to meet the expectation constraint. For comparison reasons, we keep the strict expectation constraint, but one could imagine a different approach; e.g. a gradual decrease from the initial 62.6% capacity factor to the target.

## 5 Concluding remarks

We have shown how to adapt modern portfolio theory to the challenge of allocating offshore wind turbines across a range of suitable wind farm regions. The constrained optimization due to the limited area makes the resulting portfolios more sensible. Limiting the number of wind farms has not been done in a wind farm portfolio context, likely because most earlier studies have not focused on offshore wind. When not using a maximum number of wind farms, our results indicate that the Norwegian Government's apparent strategy of sequentially opening new offshore regions for wind power deployment may lead to a sub-optimal final solution to the allocation problem but seemingly very close or even equal to the global solution. However, a step-by-step build-out may, after all, be a good idea, as it likely provides better intermediate portfolios on the way towards 30 GW installed capacity.

The NEZ has considerable potential for obtaining a diversified offshore wind power portfolio. Among our candidate locations, we found a clear block structure in the wind power correlation matrices (Figure 2). Spreading the wind farms across these blocks dramatically reduces the total variation. The maximum potential diversification effect we found was 50.4% for a capacity factor of 59.8%. From the sequential build-out case, we found that the diversification effect was also large after including only a few locations (5-7).

Deciding where to place wind farms and how to construct an investment portfolio are two different problems. We have highlighted some of these fundamental differences and how these can be taken into account or can be seen in the results of a modern portfolio analysis. The U-turning efficient frontier could also occur for financial investment portfolios, but in such applications, a risk-free interest rate is usually included in the pool of assets. A consequence of the U-turn is a unique minimum standard deviation portfolio that is not an extremity in the expected output variable.

The twenty areas that Norwegian Water Resources and Energy Directorate (2023) have identified seem reasonable from our perspective. Using the wind power suitability scores of Solbrekke and Sorteberg (2023) to identify 19 locations and adding Utsira Nord resulted in a similar pool of candidate locations, with some distinctions. The performance of the resulting portfolios for the two candidate location sets was satisfactory. The NVE portfolios did slightly better for low variance portfolios and S&S slightly better for high capacity factor portfolios in the case with the fewest restrictions. In any case, the differences are minor

and indicate that our results are robust against the selection of candidate locations.

We do not expect the Norwegian Government to decide where to place each turbine in NEZ solely based on this work. Given the points already discussed, this paper merely contributes to the discussion regarding the spatial planning of offshore wind farms in NEZ. Getting reliable cost estimates for building and maintaining offshore wind farms at different locations should be part of such a decision. It could also be that the parameters we have optimized the portfolios under do not correspond with

the risk aversion of the decision-makers. Accepting a more volatile wind farm portfolio with lower infrastructure investments seems a reasonable compromise.

*Code and data availability.*    The NORA3-WP data (Solbrekke and Sorteberg, 2022) is available at https://archive.sigma2.no/pages/public/datasetDetail.jsf?id=10.11582/2021.00068. The necessary R code and extracted NORA3-WP data for the 40 locations are published at https://github.com/holleland/OffshoreWindPortfolios in agreement with Solbrekke and Sorteberg (2022).

*Author contributions.*    SH prepared the data, developed the code and analysis, and wrote the original draft with IMS. GDB, HO and IMS contributed to conceptualization, choice of methodology, and writing. The order of the coauthors is alphabetical by surname.

*Competing interests.*    The authors declare that they have no conflict of interest.

*Acknowledgements.*    The Norwegian Research Council supported this work through the Centre for Research-Based Innovation *Climate Futures*, project number 309562.

**Appendix: Tables**



**Table A1.** Number of turbines per location for the portfolio cases (A-E) and target capacities (58, 60, 62%), with NVE locations at the top and S&S below. The far right column contains the maximum turbine constraints per location.

| | $T_{CF}$ | 58% | | | 60% | | | | | 62% | | | | | Max |
|---|---|---|---|---|---|---|---|---|---|---|---|---|---|---|---|
| | Case | A | B | C | A | B | C | D | E | A | B | C | D | E | Turb. |
| N | Nordavind A | 333 | 479 | 468 | 269 | 477 | 416 | 266 | | 150 | | 319 | 135 | | 998 |
| | Nordavind B | 125 | | | 146 | | | 146 | 522 | 174 | 523 | | 176 | 523 | 523 |
| | Nordavind C | 118 | | | 100 | | | 100 | | 67 | | | 63 | | 246 |
| | Nordavind D | 266 | 461 | 457 | 208 | 463 | 381 | 205 | | 102 | | 276 | 89 | | 850 |
| NW | Nordvest A | 204 | | | 204 | | 309 | 202 | | 148 | | | 128 | | 2639 |
| | Nordvest B | 20 | | | 41 | | | 35 | | | | | | | 802 |
| | Nordvest C | 296 | 451 | 443 | 102 | | | 101 | 486 | | | | | | 1303 |
| W | Vestavind A | | | | 194 | 407 | 368 | 208 | | 328 | 440 | 410 | 310 | 440 | 440 |
| | Vestavind B | 56 | | | 37 | | | 11 | | | | | | | 697 |
| | Vestavind C | | | | | | | | | | | | | | 243 |
| | Vestavind D | 169 | | | | | | | | | | | | | 169 |
| | Vestavind E | | | | 63 | | | | | | | | | | 345 |
| | Vestavind F | 74 | 300 | 349 | | 158 | | 100 | 279 | | 183 | | 100 | 183 | 465 |
| SE | Sørvest A | | | | 25 | | | 12 | | 43 | | | | | 340 |
| | Sørvest B | | | | | | | | | | | | | | 509 |
| | Sørvest C | | | | | | | | | | | | | | 413 |
| | Sørvest D | 159 | | | 218 | | | 160 | | 138 | | | 132 | | 284 |
| | Sørvest E | | | | 32 | | | | | 170 | | | 152 | | 238 |
| | Sørvest F | | 309 | | 28 | 495 | | 133 | 304 | 170 | 361 | 470 | 188 | 361 | 631 |
| SW | Sønnavind A | 180 | | 283 | 333 | | 526 | 321 | 409 | 510 | 493 | 525 | 527 | 493 | 677 |
| N | 1 | 229 | 424 | 399 | 191 | 436 | 386 | 180 | | 109 | | | 99 | | 500 |
| | 2 | 171 | | | 171 | | | 146 | | 161 | | 459 | 159 | | 500 |
| | 3 | 4 | | | | | | | | | 500 | | | | 500 |
| | 4 | | | | 68 | | | 129 | 500 | 165 | | | 176 | 500 | 500 |
| | 5 | 34 | | | | | | | | | | | | | 500 |
| | 6 | 311 | 429 | 434 | 246 | 404 | 352 | 213 | | 79 | | | 61 | | 500 |
| NW | 7 | 176 | | | 194 | | 316 | 179 | | 153 | | 265 | 136 | | 500 |
| | 8 | 8 | | | 72 | | | 100 | 401 | | | | | | 500 |
| | 9 | 221 | 408 | 416 | 25 | | | | | | | | | | 500 |
| W | 10 | 143 | | | 232 | 481 | 446 | 222 | 408 | 291 | 500 | 405 | 280 | 500 | 500 |
| | 11 | 45 | | 348 | | | | | | | | | | | 500 |
| | 12 | | | | | | | | | | | | | | 500 |
| | 13 | | | | 14 | | | | | | | | | | 500 |
| | 14 | 81 | 268 | | | 179 | | 100 | 191 | | 218 | | 100 | 262 | 500 |
| | 15 | 105 | | | 155 | | | 128 | | 193 | | | 160 | | 500 |
| SE | 16 | 91 | 471 | | 137 | 500 | | 167 | 500 | 177 | 297 | 371 | 167 | 280 | 500 |
| | 17 | 43 | | | 112 | | | 50 | | 162 | | | 124 | | 500 |
| | 18 | 87 | | 403 | 204 | | 500 | 221 | | 451 | 485 | 500 | 495 | 458 | 500 |
| SW | 19 | 65 | | | 157 | | | 165 | | 59 | | | 43 | | 500 |
| | 20 | 186 | | | 22 | | | | | | | | | | 500 |



**Table A2.** Number of turbines for each iteration of the sequential build-out in case D for the different targets (60, 62%) and candidate location set NVE.

| Set – $T_{CF}$ | Installed Power (GW) | 13 | 19 | 7 | 2 | 20 | 4 | 5 | 8 | 1 | 17 | 3 | 6 | 9 | 14 |
|---|---|---|---|---|---|---|---|---|---|---|---|---|---|---|---|
|  | 3 | 100 | 100 |  |  |  |  |  |  |  |  |  |  |  |  |
|  | 4.5 | 100 | 102 | 98 |  |  |  |  |  |  |  |  |  |  |  |
|  | 6 | 100 | 133 | 98 | 70 |  |  |  |  |  |  |  |  |  |  |
|  | 7.5 | 100 | 133 | 101 | 135 | 31 |  |  |  |  |  |  |  |  |  |
|  | 9 | 100 | 133 | 101 | 135 | 71 | 60 |  |  |  |  |  |  |  |  |
|  | 10.5 | 100 | 133 | 101 | 135 | 104 | 76 | 52 |  |  |  |  |  |  |  |
|  | 12 | 100 | 133 | 101 | 135 | 113 | 102 | 61 | 55 |  |  |  |  |  |  |
|  | 13.5 | 100 | 133 | 101 | 135 | 142 | 102 | 61 | 66 | 60 |  |  |  |  |  |
|  | 15 | 100 | 133 | 101 | 135 | 154 | 114 | 77 | 81 | 85 | 19 |  |  |  |  |
| NVE/60% | 16.5 | 100 | 133 | 101 | 135 | 171 | 114 | 91 | 95 | 102 | 34 | 23 |  |  |  |
|  | 18 | 100 | 133 | 101 | 135 | 187 | 118 | 107 | 110 | 122 | 49 | 39 |  |  |  |
|  | 19.5 | 100 | 133 | 101 | 135 | 203 | 129 | 122 | 124 | 142 | 64 | 47 |  |  |  |
|  | 21 | 100 | 133 | 101 | 135 | 219 | 141 | 138 | 138 | 161 | 78 | 56 |  |  |  |
|  | 22.5 | 100 | 133 | 101 | 135 | 236 | 152 | 152 | 152 | 181 | 93 | 64 | 2 |  |  |
|  | 24 | 100 | 133 | 101 | 135 | 253 | 162 | 162 | 165 | 199 | 109 | 72 | 9 |  |  |
|  | 25.5 | 100 | 133 | 101 | 135 | 270 | 173 | 172 | 178 | 218 | 124 | 80 | 15 | 1 |  |
|  | 27 | 100 | 133 | 101 | 135 | 287 | 183 | 182 | 188 | 236 | 136 | 88 | 22 | 4 | 4 |
|  | 28.5 | 100 | 133 | 101 | 139 | 304 | 194 | 192 | 198 | 252 | 148 | 95 | 28 | 8 | 8 |
|  | 30 | 100 | 133 | 101 | 146 | 321 | 205 | 202 | 208 | 266 | 160 | 100 | 35 | 11 | 12 |

| Set – $T_{CF}$ | Installed Power (GW) | 13 | 19 | 2 | 20 | 8 | 5 | 4 | 1 | 17 | 18 | 3 |
|---|---|---|---|---|---|---|---|---|---|---|---|---|
|  | 3 | 100 | 100 |  |  |  |  |  |  |  |  |  |
|  | 4.5 | 100 | 142 | 58 |  |  |  |  |  |  |  |  |
|  | 6 | 100 | 142 | 103 | 55 |  |  |  |  |  |  |  |
|  | 7.5 | 100 | 142 | 112 | 78 | 69 |  |  |  |  |  |  |
|  | 9 | 100 | 142 | 112 | 131 | 73 | 42 |  |  |  |  |  |
|  | 10.5 | 100 | 142 | 112 | 183 | 91 | 42 | 30 |  |  |  |  |
|  | 12 | 100 | 153 | 112 | 220 | 113 | 42 | 38 | 23 |  |  |  |
|  | 13.5 | 100 | 153 | 112 | 244 | 128 | 46 | 48 | 35 | 33 |  |  |
|  | 15 | 100 | 153 | 112 | 267 | 144 | 54 | 59 | 49 | 37 | 25 |  |
| NVE/62% | 16.5 | 100 | 153 | 112 | 294 | 161 | 60 | 59 | 59 | 48 | 40 | 14 |
|  | 18 | 100 | 153 | 112 | 321 | 178 | 68 | 59 | 72 | 59 | 55 | 25 |
|  | 19.5 | 100 | 153 | 116 | 347 | 194 | 76 | 59 | 82 | 70 | 70 | 34 |
|  | 21 | 100 | 153 | 124 | 373 | 211 | 83 | 59 | 90 | 81 | 84 | 42 |
|  | 22.5 | 100 | 153 | 133 | 399 | 227 | 91 | 63 | 97 | 92 | 99 | 46 |
|  | 24 | 100 | 157 | 142 | 425 | 244 | 98 | 68 | 105 | 101 | 111 | 50 |
|  | 25.5 | 100 | 165 | 150 | 450 | 260 | 105 | 73 | 112 | 109 | 122 | 53 |
|  | 27 | 100 | 173 | 159 | 476 | 277 | 113 | 79 | 120 | 116 | 132 | 56 |
|  | 28.5 | 100 | 180 | 168 | 501 | 294 | 120 | 84 | 127 | 124 | 142 | 60 |
|  | 30 | 100 | 188 | 176 | 527 | 310 | 128 | 89 | 135 | 132 | 152 | 63 |





**Table A3.** Number of turbines for each iteration of the sequential build-out in case D for the different targets (60, 62%) and candidate location set S&S.

| Set – $T_{CF}$ | Installed Power (GW) | 13 | 19 | 7 | 2 | 20 | 4 | 5 | 8 | 1 | 17 | 3 | 6 | 9 | 14 |
|---|---|---|---|---|---|---|---|---|---|---|---|---|---|---|---|
| | | 14 | 16 | 8 | 4 | 10 | 1 | 19 | 18 | 6 | 7 | 2 | 15 | 17 | |
| | 3 | 100 | 100 | | | | | | | | | | | | |
| | 4.5 | 100 | 100 | 100 | | | | | | | | | | | |
| | 6 | 100 | 129 | 100 | 71 | | | | | | | | | | |
| | 7.5 | 100 | 132 | 100 | 109 | 59 | | | | | | | | | |
| | 9 | 100 | 160 | 100 | 109 | 74 | 57 | | | | | | | | |
| | 10.5 | 100 | 167 | 100 | 109 | 87 | 74 | 64 | | | | | | | |
| | 12 | 100 | 167 | 100 | 129 | 93 | 113 | 74 | 24 | | | | | | |
| | 13.5 | 100 | 167 | 100 | 129 | 100 | 113 | 74 | 62 | 55 | | | | | |
| | 15 | 100 | 167 | 100 | 129 | 112 | 113 | 74 | 92 | 71 | 42 | | | | |
| S&S/60% | 16.5 | 100 | 167 | 100 | 129 | 128 | 113 | 75 | 114 | 79 | 56 | 39 | | | |
| | 18 | 100 | 167 | 100 | 129 | 128 | 113 | 83 | 117 | 94 | 69 | 53 | 47 | | |
| | 19.5 | 100 | 167 | 100 | 129 | 136 | 113 | 94 | 135 | 110 | 84 | 71 | 61 | | |
| | 21 | 100 | 167 | 100 | 129 | 148 | 118 | 104 | 153 | 125 | 98 | 86 | 72 | | |
| | 22.5 | 100 | 167 | 100 | 129 | 160 | 128 | 114 | 168 | 140 | 112 | 96 | 82 | 5 | |
| | 24 | 100 | 167 | 100 | 129 | 173 | 138 | 124 | 178 | 155 | 125 | 106 | 91 | 14 | |
| | 25.5 | 100 | 167 | 100 | 129 | 185 | 149 | 134 | 189 | 169 | 139 | 116 | 100 | 23 | |
| | 27 | 100 | 167 | 100 | 129 | 197 | 159 | 144 | 200 | 184 | 152 | 126 | 109 | 32 | |
| | 28.5 | 100 | 167 | 100 | 129 | 209 | 170 | 155 | 211 | 198 | 166 | 136 | 118 | 41 | |
| | 30 | 100 | 167 | 100 | 129 | 222 | 180 | 165 | 221 | 213 | 179 | 146 | 128 | 50 | |
| | | 14 | 16 | 4 | 18 | 10 | 7 | 2 | 15 | 1 | 6 | 19 | 17 | | |
| | 3 | 100 | 100 | | | | | | | | | | | | |
| | 4.5 | 100 | 140 | 60 | | | | | | | | | | | |
| | 6 | 100 | 140 | 113 | 47 | | | | | | | | | | |
| | 7.5 | 100 | 140 | 117 | 65 | 78 | | | | | | | | | |
| | 9 | 100 | 140 | 117 | 115 | 83 | 45 | | | | | | | | |
| | 10.5 | 100 | 140 | 117 | 158 | 96 | 45 | 44 | | | | | | | |
| | 12 | 100 | 140 | 117 | 185 | 106 | 53 | 68 | 32 | | | | | | |
| | 13.5 | 100 | 140 | 117 | 223 | 119 | 57 | 68 | 44 | 32 | | | | | |
| | 15 | 100 | 140 | 117 | 262 | 134 | 63 | 68 | 58 | 44 | 14 | | | | |
| S&S/62% | 16.5 | 100 | 140 | 117 | 296 | 149 | 68 | 79 | 70 | 49 | 19 | 13 | | | |
| | 18 | 100 | 140 | 117 | 302 | 163 | 75 | 91 | 77 | 55 | 26 | 18 | 36 | | |
| | 19.5 | 100 | 140 | 118 | 328 | 178 | 83 | 102 | 88 | 61 | 33 | 21 | 49 | | |
| | 21 | 100 | 140 | 126 | 353 | 193 | 90 | 111 | 99 | 66 | 37 | 24 | 62 | | |
| | 22.5 | 100 | 140 | 135 | 378 | 207 | 98 | 119 | 110 | 72 | 41 | 27 | 74 | | |
| | 24 | 100 | 140 | 143 | 402 | 222 | 106 | 127 | 120 | 77 | 45 | 31 | 87 | | |
| | 25.5 | 100 | 140 | 151 | 427 | 237 | 113 | 135 | 131 | 82 | 49 | 34 | 100 | | |
| | 27 | 100 | 149 | 160 | 450 | 251 | 121 | 143 | 141 | 88 | 53 | 37 | 108 | | |
| | 28.5 | 100 | 158 | 168 | 473 | 266 | 129 | 151 | 150 | 93 | 57 | 40 | 116 | | |
| | 30 | 100 | 166 | 176 | 495 | 280 | 136 | 159 | 160 | 99 | 61 | 43 | 124 | | |





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
