# Peer review of "Optimal allocation of 30GW offshore wind power in the Norwegian Economic Zone"

_Wind Energy Science, 2024_

## Author Response (AR1)

**Review response for**
**Optimal allocation of 30GW offshore wind power in the Norwegian Economic Zone**
**in Wind Energy Science**

Sondre Hølleland, Håkon Otneim, Geir D. Berentsen and Ida Marie Solbrekke

April 2024

**1 Reviewer 1 comments**

In the paper "Optimal allocation of 30GW offshore wind power in the Norwegian Economic Zone", the authors present and interesting and timely study of how geographical smoothening can reduce the variability of aggregate offshore wind generation in future scenarios. However, I have several questions and comments about the modeling choices and the novelty of the presented analysis.

We appreciate Matti Koivisto's comments and valuable input, which we believe has substantially improved the article. We are thankful for the opportunity to clarify and give reasons for some of the choices made in the article in question. We respond to your comments in due order.

1. In the literature review part of the Introduction section, the authors compare the applied dataset to ERA-Interim, saying that: "In this study, we use the high-resolution wind power reanalysis NORA3-WP (see section 2.1) for our analysis. In contrast with ERA-Interim, which Tejeda et al. (2018) employed, NORA3-WP has a higher temporal resolution (hourly versus 6- hourly) and a longer history (24 years versus 10 years)". However, ERA-Interim is an old dataset, which was quite some time ago replaced by ERA5. ERA5 has hourly resolution and 40+ years long history. Thus, I think the mentioned benefits of hourly resolution and long history are not quite correct, as ERA5 also has hourly resolution and actually a longer history.

   - We appreciate your comment. Indeed, the ERA-Interim is an old dataset replaced by ERA5. The study by Tejeda (2018, https://doi.org/10.1002/we.2153), a comparable study to ours, used ERA-interim. Hence, one of the advances in this study is using a more updated data set with increased temporal and spatial resolution (NORA3-WP). NORA3-WP is a wind resource and wind power data set based on NORA3. NORA3 is a dynamical downscaling of ERA5. It is correct that ERA5 has hourly resolution and a longer history (actually back to 1940). However, NORA3 and NORA3-WP have a spatial resolution corresponding to approximately 1/10 of ERA5. Improvements of NORA3 over ERA5 are quantified by Haakenstad et al. (2021, https://doi.org/10.1175/JAMC-D-21-0029.1) and Solbrekke and Sorteberg (2022, https://doi.org/10.5194/wes-6-1501-2021). We have rephrased the quoted sentence to *"In this study, we use the high-resolution wind power reanalysis NORA3-WP (see section 2.1), which is a dynamic downscaling of the state-of-the-art reanalysis ERA5 (Hersbach et al., 2020). Comparing NORA3-WP to the dataset used by Tejeda et al. (2018) NORA3-WP has a higher temporal resolution (hourly versus 6-hourly) and a more extended history (24 years versus 10 years)."* to emphasize the benefits of using NORA3, and the advances from the study by Tejeda et al (2018, https://doi.org/10.1002/we.2153).

2. Considering the comment above and the fact that modern portfolio theory has been applied extensively in analyzing wind generation variability (as the authors state), the novelty of the paper is not very clear to me. The wind data are not in any obvious way better than other available datasets, and the methodology is not new. Can you clarify what is the novelty of the presented analyses?

   - This is a crucial point because we judge the fact that the methodology is well-established and has been applied to similar problems in the past as a strength rather than a weakness.

We advance the literature by introducing new cardinality constraints in this setting and a novel sequential build-out routine, which is lacking in previous studies and we believe it will improve future studies. A sentence highlighting this has been added to the introduction of the paper. Moreover, this is the first paper to apply this methodology in the NEZ; both in the regions where Norway plans to build offshore wind farms and in regions selected using Solbrekke and Sorteberg's suitability scores. Thus, the paper informs the current debate in Norway by providing valuable insights into the diversification potential in the NEZ. The diversification is robust to these changes in candidate locations. Regarding improvements in the meteorological data set used, see the comment above.

3. Are the NORA3-WP wind speed data accurate in modelling wind speeds in the applied region? Has it been benchmarked, e.g., to ERA5 or New European Wind Atlas mesoscale runs, or other wind speed datasets? What is the reason to choose to use NORA3-WP?

- NORA3 has been validated for the region in question and also compared to ERA5. The dynamical downscaling of ERA5 resulting in NORA3 is found to be an improvement over the host data set (ERA5). Please see the studies Haakenstad et al. (2021, https://doi.org/10.1175/JAMC-D-21-0029.1) and Solbrekke and Sorteberg (2022, https://doi.org/10.5194/wes-6-1501-2021) for details. When it comes to the comparison of NORA3 to similar wind atlases, like the New European Wind Atlas (NEWA), a comparison by Cheynet et al. (2022, https://doi.org/10.1088/1742-6596/2362/1/012009) shows that NORA3 performs better than NEWA. For clarification, we have added some of the validation and comparison results under Section 2.1.

4. Is minimizing hourly resolution variability sufficient to find the "optimal" portfolio? Prolonged low/high wind events (e.g., daily, or weekly) could be more critical to the system than hourly resolution variability. Do you a) think that consideration of more prolonged events can be meaningful; and b) do you think it would make difference to the results?

- When performing a portfolio analysis, the object to be minimized has to be set, and in this study, we chose the hourly wind variability to exploit the high temporal resolution in the data set. However, other objectives can also be selected, e.g., minimize the length of a prolonged low wind event, minimize the risk of having a low wind event longer than x days, etc. We have added a subsection 3.1 "Choice of temporal scale" where we argue for why we have chosen to use hourly time resolution. We have also added a sensitivity analysis to the choice of temporal scale, based on your comment, in Appendix A. Here we consider using daily, weekly or monthly temporal scales as well as hourly. Aggregating from hourly scale will increase the dependency, and therefore the correlation between wind power production sites, and hence require larger distances between wind farms to obtain diversification effects. The resulting correlation structure for hourly and daily are qualitatively very similar but at weekly and monthly scale, the diversification potential is significantly reduced for the Norwegian offshore wind power portfolio. The appendix includes a figure illustrating the difference between the correlation matrices for the NVE regions across the different temporal scales and a table showing the distribution of turbines across the five different NVE regions. The specific choice of temporal scale may depend on the decision maker. Since the prolonged wind events are tied to long-lasting high-pressure situations over the site in question (see Solbrekke et al, 2020, https://doi.org/10.5194/wes-5-1663-2020), the diversification effect would still be highest for wind parks far apart. Hence, to some degree, the result will still be valid, choosing to minimize prolonged low wind events instead of hourly wind variability.

5. It is not clear to me how wake losses are modelled (sorry if I missed that). Please clarify. When going towards the 30 GW of installations, I would assume that individual candidate locations/areas could see quite a lot of installations, and the wake losses could become large; please clarify how this is considered.

- Wake losses are important, indeed. Both internally in a wind farm, but also inter-park wake losses. Starting with the inter-park wake losses: The Norwegian economic zone (NEZ) is so large that an installation of 30GW of offshore wind only requires approximately 1% of NEZ (Solbrekke, 2022, https://hdl.handle.net/11250/3011268). Therefore, this study handles inter-park wake losses by requiring a certain distance between the

wind parks. This study does not model the internal wind park wake losses, as it aims to find an optimal location for the whole wind farm and not every single turbine within each wind farm. In addition, the diversification effect is not influenced by a decreased capacity factor due to internal wake losses. The diversification effect is mainly obtained by large-scale differences in weather. We added a paragraph on energy system losses, including specifically wake losses, in the underlying assumptions section.

6. Why is the variability minimized only considering offshore wind Norway? Would it not be more meaningful to minimize the variability over all nearby countries and considering both onshore and offshore wind (and maybe even solar PV)? As the electricity markets are interconnected, generation is imported and exported between countries, which should allow even larger geographical area to be used to decrease the variability of the aggregate wind generation. (even if only Norwegian locations are considered as the potential locations, it would seem reasonable for me to try to minimize wind generation variability, e.g., in all of Northern Europe – which could push the optimal locations to be in further north in Norway)

   - Thank you for this interesting comment. The idea behind an isolated offshore wind power study was motivated by the recent goal of the Norwegian government to open 30GW of offshore wind by 2040 (https://www.regjeringen.no/en/aktuelt/ambitious-offshore-wind-power-initiative/id2912297/). Thus, this study looks at the potential to exploit all NEZ when carrying out this plan. In addition to that, in 2020, the Norwegian Government opened two offshore areas for large-scale wind power applications (*Sørlige Nordsjø 2* and *Utsira Nord*), and they decided that no European cable for export will be built at this stage, and all the electricity will go back to the Norwegian mainland. Including other renewable energy sources would be highly interesting, especially hydro power, but that was not the scope of this study. It has definitely potential to be a future study.

7. In the section "Underlying assumptions", the authors explain that transmission line limits and the broader power and energy system are left outside of the scope of the paper. However, it begs the question if optimal portfolio theory is the most appropriate method for the analysis? E.g., an energy system analysis tool (e.g., PyPSA, Balmorel, PLEXOS, or similar) would naturally allow both transmission lines and the wider flexibility in the energy system (hydropower, batteries, demand response, etc.) to be included in the analysis. Would it not be a better approach to find the optimal installation locations? Please discuss the pros and cons of these two widely used approaches.

   - Thank you for this comment. There are indeed pros and cons to both approaches. As we have established in your comments above, the MPT methodology is a standard approach for solving this type of problem. However, we agree that it would be interesting to use a tool that can take into account the limitations and structure of the energy system. This would certainly increase the complexity of the analysis. You are left with two alternatives: 1) Assume status quo for the energy system, which is unrealistic, or 2) make some assumptions on how the future energy system will be. It is clear that increasing the installed power by 30 GW will have a large impact on the broader power- and energy system, both in terms of energy in the system and the price of electricity in connected areas. We argue that the power system as it is today may not be relevant 30 years from now. Comprehensive plans for new transmission lines, both internally in Norway and across borders, exist (see e.g. https://www.statnett.no/globalassets/for-aktorer-i-kraftsystemet/planer-og-analyser/atk/analyse-av-transportkanaler-2023-2050.pdf, in Norwegian) and new energy sources may be built during this time period. Focusing exclusively on offshore wind power enables us to answer some relevant questions using a simple and well-established methodology, allowing others to add more details later when more information is available.

**2   Reviewer 2 comments**

**General Comments**

Thank you for the opportunity to review the study. The paper is well written and provides a valuable initial verification piece to the assessment conducted by NVE. As mentioned at line 328, the analysis could be strengthened by considering offshore wind spatial drivers such as investment

and maintenance costs not just power production. Another consideration may consider technology or innovation advancements in wind turbine and installation strategies for offshore wind projects as we approach 2030 and beyond. The study would benefit further review by an economist since the framework considers an investment portfolio assessment.

Thank you for your comments. We appreciate the anonymous reviewer's comments and valuable input.

Technology and innovation are certainly something to consider and mention. We have added a sentence about this in the underlying assumptions section 3.3 with specific reference to wake losses, as it was well-suited there: *"Technology and innovation advancements in wind turbine and installation strategies may also reduce the effect of wake losses as we approach 2030 and beyond."* More on this in the first response below.

**Specific Comments**

- Table 1: Mean CF of 65.6% seems high, especially if this value is net CF. May be worth clearly specifying whether the CF values consider wind energy system losses; if so, please clearly state the loss assumptions.

  We do not explicitly take into account energy system loss here. The reported 65.6% CF is a theoretical CF purely based on wind speed and the power curve of said turbine. This is what you would expect from placing one turbine at that location. Introducing energy system losses (such as electrical resistance losses, converter losses, maintenance losses, wake losses and auxiliary power consumption) would complicate the analysis further. We argue that many of these effects are common for all wind farm locations with the exception of electrical resistance losses, where the energy lost in underwater cables depends on the length of the cable. This is accounted for in the selection of potential wind farm locations identified based on Solbrekke and Sorteberg (2022), as their suitability score penalize distance to shore. However, this is only covered implicitly in our work and not part of the turbine placement once a candidate location has been identified. Taking into account wake losses could be an interesting extension, but would depend heavily on assumptions regarding how the turbines are placed within a wind farm, which is outside the scope of our article.

  To make this more clear, we have added the following paragraph to section 3.3 Underlying assumptions: *"We do not explicitly take into account any energy system losses, such as electrical resistance losses, converter losses, maintenance losses, wake losses and auxiliary power consumption. Electrical resistance losses depend on the length of the cable. For the S&S-locations, distance to shore is penalized in calculating the wind power suitability score. Hence, this is part of the decision of selecting candidates. It is, however, not part of the placing of turbines. Wake losses are also important and would affect the total production from a wind farm, especially the larger ones (Barthelmie et al., 2009; Ghaisas et al., 2017). Quantifying wake losses would involve making assumptions about how the turbines are placed within a wind farm, but we focus on a more macro level. Our analysis, however, concerns the distribution of turbines across wind farms. Technology and innovation advancements in wind turbine and installation strategies may also reduce the effect of wake losses as we approach 2030 and beyond."*

- Figure 3: Recommend adding a note or reference for the "Diversification effect: 50.4%" called out in the figure name. It looks like this is explained in a bit of detail on line 349 but may be worth including an explanation with the other details of Figure 3.

  Thank you for this comment. We agree that the figure captions should be self-contained, and have added the following sentence to the figure caption explaining what the blue arrow indicates: *"The blue arrow indicates a 50.4% reduction in portfolio standard deviation from placing all turbines at Vestavind F to the minimum standard deviation portfolio of the NVE locations for Case A with $T_{CF} = 59.8\%$."*

- Table 3: Perhaps this was overlooked, but it's difficult to find the GW in parenthesis mentioned to be included in the table.

  You are correct. This was something we had included in an earlier version of the table, but later removed - forgetting to update the caption. The caption has now been corrected accordingly. Thank you for pointing this out.

**Technical Corrections**

- Line 470: Last sentence may need to be edited '. . . seems like a. . . ' or similar.

  Thank you for this comment. We have changed the final sentence to: *"Accepting a more volatile wind farm portfolio with lower infrastructure investments seems like a reasonable compromise."*